

# Acute ventilatory responses to swimming at increasing intensities

Ana Sofia Monteiro[1,2], José Francisco Magalhães[1,2], Beat Knechtle[3,4], Cosme F. Buzzachera[5], J. Paulo Vilas-Boas[1,2] and Ricardo J. Fernandes[1,2]

[1] Faculty of Sport of University of Porto, Centre of Research, Education, Innovation and Intervention in Sport, Porto, Portugal
[2] Faculty of Sport of University of Porto, Porto Biomechanics Laboratory, Porto, Portugal
[3] University of Zurich, Institute of Primary Care, Zurich, Switzerland
[4] Medbase St. Gallen am Vadianplatz, St. Gallen, Switzerland
[5] University of Pavia, Department of Public Health, Experimental and Forensic Medicine, Pavia, Italy

Corresponding author
Ricardo J. Fernandes,
ricfer@fade.up.pt

## ABSTRACT

**Background:** Physical exercise is a source of stress to the human body, triggering different ventilatory responses through different regulatory mechanisms and the aquatic environment imposes several restrictions to the swimmer, particularly regarding the restricted ventilation. Thus, we aimed to assess the acute ventilatory responses and to characterize the adopted breathing patterns when swimming front crawl at increasing intensity domains.

**Methods:** Eighteen well-trained swimmers performed $7 \times 200$ m front crawl ($0.05$ m·s$^{-1}$ velocity increments) and a maximal 100 m (30 s rest intervals). Pulmonary gas exchange and ventilation were continuously measured (breath-by-breath) and capillary blood samples for lactate concentration ([La$^-$]) analysis were collected at rest, during intervals and at the end of the protocol, allowing the identification of the low, moderate, heavy, severe and extreme intensity domains.

**Results:** With the swimming velocity rise, respiratory frequency ($f_R$), [La$^-$] and stroke rate (SR) increased ([29.1–49.7] breaths·min$^{-1}$, [2.7–11.4] mmol·L$^{-1}$, [26.23–40.85] cycles; respectively) and stroke length (SL) decreased ([2.43–2.04] m·min$^{-1}$; respectively). Oxygen uptake (VO$_2$), minute ventilation (VE), carbon dioxide production (VCO$_2$) and heart rate (HR) increased until severe ([37.5–53.5] mL·kg$^{-1}$·min$^{-1}$, [55.8–96.3] L·min$^{-1}$, [32.2–51.5] mL·kg$^{-1}$·min$^{-1}$ and [152–182] bpm; respectively) and stabilized from severe to extreme ($53.1 \pm 8.4$, mL·kg$^{-1}$·min$^{-1}$, $99.5 \pm 19.1$ L·min$^{-1}$, $49.7 \pm 8.3$ mL·kg$^{-1}$·min$^{-1}$ and $186 \pm 11$ bpm; respectively) while tidal volume (V$_T$) was similar from low to severe ([2.02–2.18] L) and decreased at extreme intensities ($2.08 \pm 0.56$ L). Lastly, the $f_R$/SR ratio increased from low to heavy and decreased from severe to the extreme intensity domains ($1.12 \pm 0.24$, $1.19 \pm 0.25$, $1.26 \pm 0.26$, $1.32 \pm 0.26$ and $1.23 \pm 0.26$).

**Conclusions:** Our findings confirm a different ventilatory response pattern at extreme intensities when compared to the usually evaluated exertions. This novel insight helps to understand and characterize the maximal efforts in swimming and reinforces the importance to include extreme efforts in future swimming evaluations.

## INTRODUCTION

Breathing is a natural and fundamental human behavior that allows the exchange of respiratory gases between the lungs and the atmosphere. When we are under stress, as physical exercise, minute ventilation ($\dot{V}E$) increases (*Pelarigo et al., 2016*; *Tipton et al., 2017*) due to the respiratory frequency ($f_R$) and tidal volume ($V_T$) rise. During an incremental exercise, $f_R$ increases non-linearly and $V_T$ tends to present a plateau, with the $\dot{V}E$ rise at lower intensities depending on both $f_R$ and $V_T$ increases. The further growth in $\dot{V}E$ at higher exercise intensities seems to be explained by the increase in $f_R$, a phenomenon known as the tachypneic breathing pattern (*Sheel & Romer, 2012*; *Nicolò, Marcora & Sacchetti, 2020*). Despite the well-established knowledge on the $f_R$ and $V_T$ contributions for $\dot{V}E$ increase during an incremental exercise, further research focusing on the different regulatory mechanisms that drive these contributions is welcome (*Figueiredo et al., 2013*; *Tipton et al., 2017*).

Central command, muscle afferent feedback and metabolic inputs are the major $\dot{V}E$ behavior determinants despite acting with different timings when exercise intensity changes (*Forster, Haouzi & Dempsey, 2012*; *Duffin, 2014*; *Tipton et al., 2017*). The regulation of $f_R$ and $V_T$ is less studied but it was previously suggested that the inputs driving $\dot{V}E$ act separately on these variables, with central command and muscle afferent feedback preferentially regulating $f_R$ (*Amann et al., 2010*; *Nicolò et al., 2017*), while metabolic responses are responsible for the $V_T$ regulation (*Nicolò et al., 2017*). Considering the great importance of the central command on $f_R$ control and the close association between breathing patterns, exercise modes and limbs movement (*Sheel & Romer, 2012*; *Forster, Haouzi & Dempsey, 2012*), it is of great importance to understand how the ventilatory response adapts to different exercise related constraints.

In swimming, the aquatic environment imposes significant restrictions on the human body, such as the increase of the hydrostatic pressure around the chest, resulting in an augmented work of the inspiratory muscles (*Lomax & McConnell, 2003*; *Leahy et al., 2019*). In addition, the swimming typical horizontal position leads to face immersion and, consequently, to restricted ventilation (*Holmér et al., 1974*; *McCabe, Sanders & Psycharakis, 2015*). These restrictions oblige swimmers to synchronize active inspiratory and expiratory phases with movements of upper and lower limbs, resulting in specific swimming breathing patterns (*Leahy et al., 2019*). Front crawl is the most common (in training and competition conditions) from the four swimming conventional techniques, with swimmers more generally inspiring on every two or three upper limbs actions, *i.e.*, using unilateral and bilateral breathing patterns (*Seifert, Chollet & Allard, 2005*; *Figueiredo et al., 2013*). Despite the existing variability of the $\dot{V}E$ responses along the different intensity domains, particularly when using the front crawl technique (*Ribeiro et al., 2015*; *Monteiro et al., 2022*), the $f_R$ and $V_T$ behaviors are still scarcely studied when swimming at increasing paces.

Since further research on the $f_R$ and $V_T$ responses is necessary to improve the overall understanding of breathing physiology and ventilatory control, we have aimed to assess the acute ventilatory responses when swimming from low to extreme intensity domains. For achieving that purpose, swimmers were required to wear a breathing snorkel attached to a gas analyzer along a standard incremental front crawl protocol. Complementarily, we aimed to characterize the swimmers breathing patterns along the exercise intensity rise to understand if the synchronization with the upper and lower limbs motion is maintained even when using the respiratory snorkel, *i.e.*, without constraining the inspiratory and expiratory phases. We have hypothesized that: (i) despite the respiratory constraints, gas exchange variables increase concomitantly with the swimming velocity rise, with $f_R$ and $V_T$ presenting a nonlinear increase and a stabilization (respectively); and (ii) swimmers keep the breathing patterns used in free swimming when breathing into a snorkel (due to the breathing synchronization with stroke rate).

## MATERIALS AND METHODS

### Participants

Eighteen well-trained swimmers (nine males) volunteered to participate in the current study. Their main anthropometric, training background and competitive characteristics were (for males and females, respectively): 20.1 ± 8.0 *vs.* 16.8 ± 1.8 years of age, 176.6 ± 7.6 *vs.* 163.4 ± 4.7 cm of body height, 67.5 ± 12.1 *vs.* 57.3 ± 6.5 kg of body mass, 21.5 ± 2.7 *vs.* 21.4 ± 1.6 kg·m$^{-2}$ of body mass index, 8.3 ± 3.8 *vs.* 7.3 ± 3.4 years of swimming practice and 489 ± 66 *vs.* 478 ± 83 Fédération Internationale de Natation points of their best competitive performance event. Participants were recruited *via* personal contact and based on the following eligibility criteria: (i) without a history of cardiorespiratory and physical diseases or injuries within the previous 6 months; (ii) having ≥2 years of swimming training background and (iii) being engaged at ≥5 training sessions per week. All the experiments were approved by the Faculty of Sport of University of Porto ethics committee (CEFADE 25 2020) and participants were informed about the purpose, benefits and any associated risks (providing their written individual consent for participation in accordance with the Helsinki Declaration).

### Experimental protocol

Subjects were asked to be rested and fully hydrated, and refrained from alcohol and caffeine consumption (and from vigorous exercise) for, at least, 24 h prior the evaluation. Test sessions were conducted in a 25 m indoor pool, with 27 °C and 26.5 °C of water and air temperatures (respectively) and 75% of humidity. After a 600 m low intensity in-water warm-up, each swimmer performed a front crawl discontinuous incremental protocol, consisting of 7 × 200 m (with 0.05 m·s$^{-1}$ velocity increments), followed by a maximal 100 m, with 30 s rest intervals in-between (adapted from *Fernandes et al., 2005*; *Carvalho et al., 2020*; *Monteiro et al., 2022*). The paces for each swimmer 7$^{th}$ step were established based on the individual 400 m front crawl performance on the evaluation day, then six velocity increments were subtracted. Swimming velocities were controlled using flashing lights on the bottom of the pool (Pacer2Swim; KulzerTEC, Aveiro, Portugal), with in-water

starts and open turns (without underwater gliding) being used due to the impossibility of performing flip turns with deep water gliding when using a respiratory snorkel.

A portable gas analysis system (K4b$^2$; Cosmed, Rome, Italy) was transported on a steel cable above the water surface allowing to measure breath-by-breath pulmonary gas exchange and ventilation by connecting the swimmer through a low hydrodynamic resistance respiratory snorkel and valve system (Aquatrainer®, Cosmed, Rome, Italy; *Ribeiro et al., 2015*). This gas analysis system was calibrated before each experimental session using ambient air against known concentrations (16% $O_2$ and 5% $CO_2$) and a 3 L calibration syringe. Heart rate (HR) was continuously recorded at the baseline and during the incremental protocol using a Polar Vantage NV (Polar Electro Oy, Kemple, Finland) that telemetrically emitted to the portable gas analyzer unit (*de Jesus et al., 2015*). Lactate concentration ([La⁻]) values were obtained using fingertip capillary blood samples collected at rest, immediately after the end of each step and at 1, 3, 5 and/or 7 min post-protocol (until obtaining maximal values) using a portable analyzer (Lactate Pro2; Arkay Inc., Kyoto, Japan; *Carvalho et al., 2020*). Stroke rate (SR) was assessed through the number of upper limbs cycles per minute in the last 50 m of each step (using a Finis stopwatch with a frequency meter function) and stroke length (SL) was calculated by dividing the mean velocity by SR (*Fernandes et al., 2005*; *Monteiro et al., 2022*).

## Data analysis

The pulmonary gas exchange and ventilation data were examined to exclude occasional errant breaths (eventually caused by swallowing, coughing or signal interruptions). For the analysis, only the values of oxygen uptake ($\dot{V}O_2$) between ±3 SD were considered (*Monteiro et al., 2020*), which were then smoothed using a three breaths moving average and 10 s time average (*Fernandes et al., 2012*). The mean values from the last 30 s of exercise per step were selected and conventional physiological criteria were applied to stablish the maximal oxygen uptake $\dot{V}O_2$max; (*Howley, Bassett & Welch, 1995*; *Zacca et al., 2020*). The lactate-velocity curve modelling method, through the determination of the interception point of the best fit of a combined linear and exponential pair of regressions, was used to determine the individual anaerobic threshold (*Carvalho et al., 2020*; *Monteiro et al., 2022*). Using the $\dot{V}O_2$max and the anaerobic threshold as physiological indicators, the following intensity domains were identified (Fig. 1): (i) the low and moderate domains, corresponding to two steps below and the step at the anaerobic threshold; (ii) the heavy and severe domains, matching the step below and the step where $\dot{V}O_2$max was elicited; and (iii) the extreme domain, allocated to the maximum 100 m at the end of the incremental protocol (*Fernandes et al., 2012*; *de Jesus et al., 2015*; *Ribeiro et al., 2017*). Swimmers breathing patterns were determined by calculating the ratio between $f_R$ and SR.

## Statistical analysis

A sample size of 18 subjects was required for a paired sample design to detect a moderately large effect size (0.83) with a 5% significance level and 95% power (G*Power 3.1.9.7; Heinrich Heine Universität Düsseldorf, Düsseldorf, Germany). Statistical procedures were conducted using SPSS (version 27.0.1.0; IBM Corporation, Armonk, NY, USA) and the

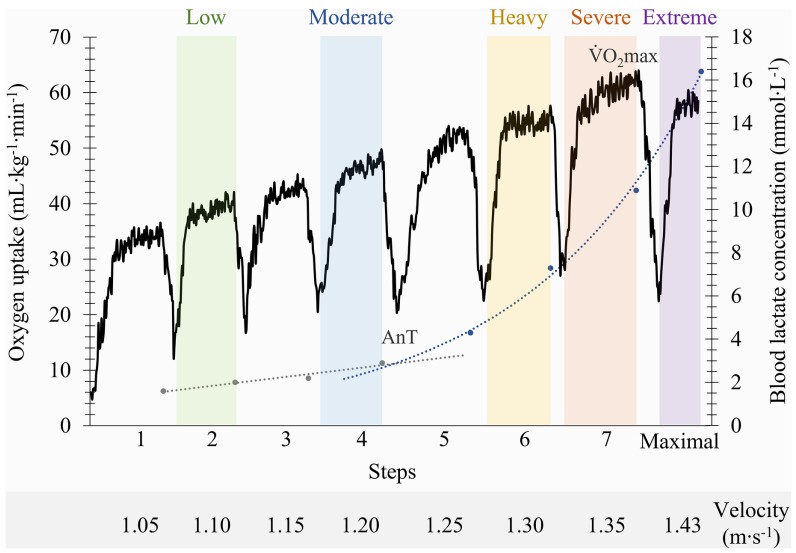

**Figure 1 Identification of the front crawl swimming intensity domains on a representative participant based on anaerobic threshold (AnT) and maximal oxygen uptake ($\dot{V}O_2max$) indicators.**

normal data distribution was checked for all variables using the Shapiro-Wilk test. Mean and standard deviation values were computed for descriptive analysis of all variables and one-way repeated measures analysis of variance with Bonferroni adjustment was used to compare the assessed physiological and performance variables along the intensity domains spectrum ($p \leq 0.05$ level). Partial eta-squared ($\eta_p^2$) for effect size calculation was computed to compare the magnitude of changes between swimming intensity domains.

## RESULTS

Depending on the swimmer, the establishment of the low and moderate intensity domains corresponded to the swimming velocity between the first-third and third-fifth steps (respectively), while the heavy, severe and extreme intensity domains corresponded to the sixth, seventh and maximal last protocol steps (in this order). The low, moderate, heavy, severe and extreme efforts were performed at $1.04 \pm 0.11$, $1.13 \pm 0.11$, $1.22 \pm 0.10$, $1.26 \pm 0.10$ and $1.39 \pm 0.11$ m·s$^{-1}$ (respectively) and all the physiological and performance variables are presented in Fig. 2. With the swimming intensity rise ($Z_{4.68} = 305.79$, $\eta_p^2 = 0.95$, $p < 0.001$), $f_R$, [La$^-$] and SR increased ($\eta_p^2 = 0.81$, $\eta_p^2 = 0.95$ and $\eta_p^2 = 0.88$, respectively; $p < 0.001$) and SL decreased ($\eta_p^2 = 0.59$, $p < 0.001$). $\dot{V}O_2$, $\dot{V}E$, $\dot{V}CO_2$ and HR increased from low to severe intensities ($\eta_p^2 = 0.91$, $\eta_p^2 = 0.88$, $\eta_p^2 = 0.89$ and $\eta_p^2 = 0.83$, respectively; $p < 0.001$), but all stabilized at extreme exertion and $V_T$ presented similar values from low to severe exertions and decreased at the extreme intensity ($p = 0.006$). The $f_R$/SR ratio increased from low to moderate ($p = 0.01$) and from moderate to heavy domains ($p = 0.02$) and lower values were observed at extreme compared to severe intensity ($p = 0.02$; Fig. 3).

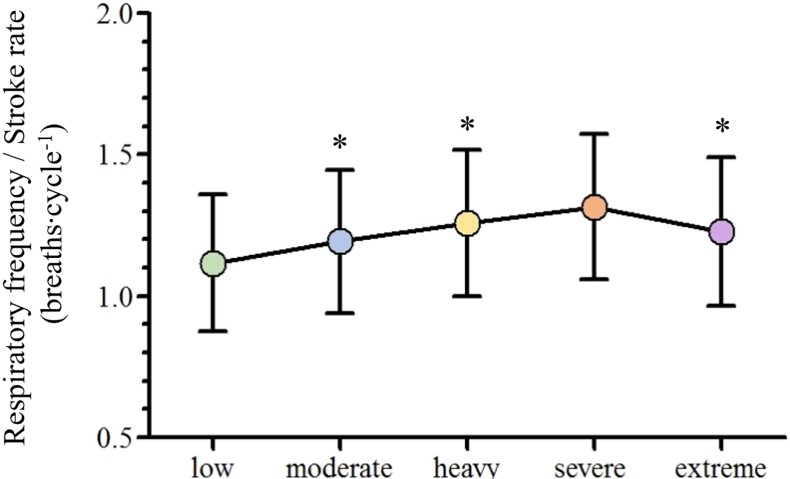

**Figure 2 Scatter plots with mean and standard deviation of the ventilatory and performance variables responses along with the swimming intensity rise.** An asterisk (*) indicates difference from the previous intensity domain ($p < 0.05$).

**Figure 3 Mean and standard deviation of the adopted breathing patterns along with the swimming intensity rise.** An asterisk (*) indicates difference from the previous intensity domain ($p < 0.05$).

## DISCUSSION

The main purpose of the current study was to assess swimmers acute ventilatory responses when performing front crawl from low to extreme intensities. As hypothesized, the values of the selected gas exchange variables increased along with the swimming intensity rise (until the severe intensity domain) and $f_R$ and $V_T$ presented a nonlinear increase and a plateau (respectively) in the incremental protocol. Concurrently, we aimed to analyze swimmers breathing pattern behavior as swimming pace was rising, being observed an $f_R$/SR ratio increase until reaching the severe intensity domain and a decrease at extreme exertions. This does not confirm our initial hypothesis that the front crawl breathing pattern was going to remain stable when swimming with a respiratory snorkel.

It is well established that the $7 \times 200$ m front crawl intermittent incremental protocol allows collecting capillary blood for [La$^-$] analysis (*Fernandes et al., 2005*; *Pelarigo et al., 2016*; *Carvalho et al., 2020*) and, together with the gas exchange assessment, ensures a complete physiological characterization of the low, moderate, heavy and severe intensity domains (*de Jesus et al., 2015*; *Zacca et al., 2019*; *Monteiro et al., 2022*). The current results are similar to those previously presented for the low-severe swimming intensity domains when using the same methodological approach, particularly regarding $\dot{V}O_2$ (*Fernandes et al., 2012*; *de Jesus et al., 2015*), $\dot{V}E$ and $f_R$ (*Pelarigo et al., 2016*; *Monteiro et al., 2022*), and [La$^-$] values (*Štrumbelj et al., 2007*; *Sousa, Vilas-Boas & Fernandes, 2014*; *Monteiro et al., 2022*).

However, most official swimming events (such as the 50, 100 and 200 m distances) occur at the extreme intensity domain, and is why a complete swimming ventilatory characterization should also include extreme efforts. This is a fundamental training zone for excelling competitive swimmers performances where the exertions are so intense that fatigue occurs and exercise ends before $\dot{V}O_2$max can be reached (*Hill, Poole & Smith, 2002*; *Ribeiro et al., 2017*). Studies focusing on the anaerobic capacity development are very scarce, with swimmers acute ventilatory responses remaining almost unexplored, justifying the inclusion of a 100 m maximal bout at the end of the front crawl incremental protocol. This maximal intensity short duration effort, when swimming up at the standard $7 \times 200$ m step protocol, allows swimmers to have their physiological profile fully characterized.

The current ventilatory results at the maximal 100 m front crawl evidenced lower $\dot{V}O_2$ and $\dot{V}E$ values compared to those reported for the same intensity domain (*Sousa, Vilas-Boas & Fernandes, 2014*), probably due to the sample characteristics (higher level and male swimmers only) and the higher swimming velocity achieved (the current 100 m bout was part of an incremental protocol instead of an isolated rectangular test). Additionally, higher $f_R$ values were observed at the current extreme intensity domain compared to a maximal 200 m front crawl bout performed at a lower swimming velocity (*Štrumbelj et al., 2007*). This demonstrates that the extreme exertions (only inferiorly delimited by the $\dot{V}O_2$max) can include a wider range of swimming velocities, being important to consider them when comparing the results obtained at this intensity domain.

Swimming faster implied a $\dot{V}O_2$, $\dot{V}E$, $f_R$, $\dot{V}CO_2$, [La$^-$] and SR increase and a SL decrease from low to severe exertions, as previously described (*Figueiredo et al., 2013*; *de Jesus et al.,*

2015; *Monteiro et al., 2022*), while $\dot{V}O_2$, $\dot{V}E$, $\dot{V}CO_2$ and HR stabilized from severe to extreme intensities (*Sousa, Vilas-Boas & Fernandes, 2014*). The attainment of $\dot{V}O_2$max at severe intensity paces, and the fact that these abovementioned variables are highly related, explain the maintenance of similar values despite the swimming intensity rise (*Sousa, Vilas-Boas & Fernandes, 2014*; *Nicolò et al., 2018*; *Monteiro et al., 2022*). HR at $\dot{V}O_2$max corresponded to 90.4% ± 3.1% of its maximum, in accordance with the secondary criteria used to confirm $\dot{V}O_2$max (*Howley, Bassett & Welch, 1995*; *Zacca et al., 2020*). Maximal [La$^-$] values were observed at extreme exertions, where the energy production is highly dependent on anaerobic metabolism, with a higher production of lactate and, consequently, its progressive accumulation in the bloodstream (*Hargreaves & Spriet, 2020*). Contrarily to what was described (*Sousa, Vilas-Boas & Fernandes, 2014*), our [La$^-$] values increase from severe to extreme intensity domains is explained by a bigger velocity rise (~10% instead of 5%), corroborating the existence of a wide range of swimming velocities at this domain.

The observed $\dot{V}E$, $f_R$ and $V_T$ behaviors along the low to extreme swimming intensity domains spectrum corroborates what is described in the literature, independently of the exercise modality performed (*Amann et al., 2010*; *Nicolò et al., 2017*). This seems to indicate that, regardless of the swimming movements and the different constraints imposed by the aquatic environment, the central command, muscle afferent feedback and metabolic inputs have the same influence on their regulation along the intensity domains spectrum (*Štrumbelj et al., 2007*; *Sheel & Romer, 2012*; *Forster, Haouzi & Dempsey, 2012*). However, the swimming intensity rise resulted in the selection of different breathing patterns at each intensity domain. Diversely to what was initially expected, the $f_R$/SR ratio tended to increase until the heavy intensity domain, indicating that swimmers took advantage of free breathing while using the respiratory snorkel (*Štrumbelj et al., 2007*). The $f_R$/SR ratio decrease from severe to extreme intensity domains can be justified by both the maximal intensity and the short time duration effort of the 100 m exertion, where SR increased more than $f_R$ (16% *vs.* 8%, respectively). In addition, the lower increase in $f_R$ compared to SR seems to indicate that this extreme effort is characterized by moments of apnea.

## CONCLUSIONS

The $f_R$ and, consequently, the $f_R$/SR ratio values were influenced by the use of the respiratory snorkel and its interpretation may be different compared to free swimming. However, this is the only methodology that provides a real time and breath-by-breath assessment of the swimmers ventilatory responses. In conclusion, by proposing the addition of a maximal effort at the end of the front crawl intermittent incremental swimming protocol, the current study provides a novel framework of the acute ventilatory responses to the large spectrum of swimming intensity domains, particularly at the extreme exertion, used both in training and competition contexts. $\dot{V}O_2$, $\dot{V}E$ and $\dot{V}CO_2$ stabilized, and $V_T$ decreased, from severe to extreme intensity domains, differently to what happened from low to severe exertions, while $f_R$ and SR increased along the swimming

intensities spectrum. The breathing pattern varied along the incremental protocol and its synchronization with stroke rate was not verified when using the respiratory snorkel.

## ACKNOWLEDGEMENTS

The authors would like to acknowledge all study participants and collaborators.

### Funding

This investigation was supported by the Portuguese Foundation for Science and Technology, I.P. (FCT) and European Union (EU) under grant number 2020.07714.BD, endorsed to Ana Sofia Monteiro. The funders had no role in study design, data collection and analysis, decision to publish, or preparation of the manuscript.

### Grant Disclosures

The following grant information was disclosed by the authors:
Portuguese Foundation for Science and Technology, I.P. (FCT) and European Union (EU): 2020.07714.BD.

### Competing Interests

Cosme F. Buzzachera is a PeerJ Academic Editor. The authors declare that they have no competing interests.

### Author Contributions

- Ana Sofia Monteiro conceived and designed the experiments, performed the experiments, analyzed the data, prepared figures and/or tables, authored or reviewed drafts of the article, and approved the final draft.
- José Francisco Magalhães performed the experiments, prepared figures and/or tables, and approved the final draft.
- Beat Knechtle conceived and designed the experiments, authored or reviewed drafts of the article, and approved the final draft.
- Cosme F. Buzzachera conceived and designed the experiments, analyzed the data, authored or reviewed drafts of the article, and approved the final draft.
- J. Paulo Vilas-Boas conceived and designed the experiments, analyzed the data, authored or reviewed drafts of the article, and approved the final draft.
- Ricardo J. Fernandes conceived and designed the experiments, performed the experiments, analyzed the data, authored or reviewed drafts of the article, and approved the final draft.

### Human Ethics

The following information was supplied relating to ethical approvals (*i.e.*, approving body and any reference numbers):

All the experiments were approved by the Faculty of Sport of University of Porto ethics committee (CEFADE 25 2020).

## Data Availability

The raw data is available in the Supplemental File.

## Supplemental Information

Supplemental information for this article can be found online at http://dx.doi.org/10.7717/peerj.15042#supplemental-information.

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
