# Peer review of "Acute ventilatory responses to swimming at increasing intensities"

_PeerJ, doi:10.7717/peerj.15042_

## Round 0.1 · original submission · Minor Revisions

Dear Authors,

Please make minor revisions according to the suggestions of reviewers or write a detailed rebuttal.

·

Basic reporting

This is a well-structured and generally very clear manuscript and the authors should eb congratulated for its conciseness. There are some grammatical errors, which I have highlighted in the comments below for the authors to address. Appropriate literature has been included and the figures included are clear and necessary.

I have suggested that one of the hypotheses (ii) be removed as the methods cited do not permit it to be examined. However, his does not detract from the overall high quality of the research undertaken and its removal will only strengthen, not weaken, the paper.

The raw data supplied is clearly and concisely formatted; although it does not contain any participant descriptive information such as age, mass, height and sex.

Experimental design

The authors have included a sound and robust method. The method has been included in sufficient detail to be replicated and an appropriate ethics statement has been included. It is clear what research question is being addressed and this research question does require investigation.

Validity of the findings

The statistics adopted are appropriate for the research question and the results have been clearly and succinctly presented. The results of the study will contribute valuable information to the swimming physiology literature.

Additional comments

This is a well-designed and executed study and the findings are worthy of publication. I have some specific comments below, which will strengthen the paper further.

Abstract, background. Please place the word snorkel as follows ‘Thus we aimed to assess the ventilatory responses and to characterize the adopted breathing patterns while swimming front crawl with a snorkel at increasing intensity domains’.
Introduction. Line 96-97. It is unclear how the experimental protocol permitted assessment of hypothesis ii ‘swimmers keep the breathing patterns used in free swimming when breathing into a snorkel (due to the breathing synchronization with stroke rate)’. It does not appear that freely breathing performance (specifically breathing frequency) was assessed at any point? Please clarify in the experimental protocol how this was assessed. Alternatively, simply remove the hypothesis - doing so would strengthen the key message.
Experimental protocol line 121. Am assuming that only one, 100 m swim was performed after the final 200 m repeat? However, this is not clear. Please clarify in the paper.
Data analysis, lines 147-150. Please address the grammatical error here. Suggest replacing with ’Data were only included for analysis if oxygen uptake values were between ± 3 SD (Monterio et al., 2020). Data were then smoothed using a three-breath rolling average and 10 s time averaging (Fernandes et al., 2012).’
Statistical analyses – it as nice see that the authors included the pre-study sample size calculations as this is often omitted in published papers.
Discussion, line 198. Unclear what is meant by posterior decrease?
Discussion, lines 201-212. Replace ‘reason’ with ‘and is’
Discussion, line 224-229. It is possible that with the use of a snorkel breathing pattern deviates from usual at maximal intensities (we have observed this in our lab during VO2max testing) permitting it to increase. Do you think that the increase in breathing frequency is simply an artefact of using a snorkel which provides a distinct advantage for oxygen uptake at higher intensities or would a similar increase also be observed during free breathing? This is worth commenting on even if the answer is 'unknown'.
Discussion, lines 227-229. Please address the grammatical errors. I’m not sure of the point being made here - is the point that higher intensities should be included in ventilatory assessments?
Discussion, line 239. Replace ‘of’ with ‘on’ and remove ‘the’.
Discussion, line 247. ‘…regardless of the…’ rather than ‘regardless the’
Discussion, lines 251-254. I agree and I think it is worth reiterating here that the addition of the snorkel could have changed breathing pattern (notably increased breathing frequency) from that occurring during freely breathing swimming. It’s a shame the authors did not include a free breathing trial as this would allow breathing frequency to be compared, and changes quantified, over the range of intensities with and without a snorkel - this would also have permitted hypothesis ii to be addressed.
Discussion, line 257. To correct the grammatical error I suggest rewording to ‘In addition, the lower increase in fr compared to SR….’
Conclusion, line 270-271. I agree and this provides further justification for removing hypothesis ii.
Figures 2 and 3 titles. Suggest replacing raise with rise.

Reviewer 2 ·

Basic reporting

The article meets the journal's standards: the language is clear and professional, literature references are complete, article structure is professional, conclusions limited to obtained results.

Experimental design

Experimental design is sufficiently described, meets the required standards.

Validity of the findings

Valid findings are clearly presented.

·

Basic reporting

In attached word document I reccomended corrections regarding writing and English language.
In general, language is clear and unambiguous and professional English used throughout the paper.
The article includes sufficient introduction and background to demonstrate how the work fits into the broader field of knowledge. Relevant prior literature is appropriately referenced. The structure of the article is in an acceptable format of ‘standard sections’. Figures are relevant to the content of the article, of sufficient resolution, and appropriately described and labeled. All appropriate raw data have been made available in accordance with our Data Sharing policy.

Experimental design

The submission clearly defines the research question. The knowledge gap being investigated is identified, and statements are made as to how the study contributes to filling that gap. Methods are described with sufficient information to be reproducible by another investigator.

Validity of the findings

Data on which the conclusions are based are provided in an acceptable repository. Data are robust, statistically sound and controlled.

Additional comments

This article is acceptable with minor language corrections (please see attached document and track changes)

---

## Round 0.2 · accepted · Accept

Dear Authors,

Your manuscript is acceptable for publication in its current form.